# Low Gut Microbial Diversity Augments Estrogen-Driven Pulmonary Fibrosis in Female-Predominant Interstitial Lung Disease

**DOI:** 10.3390/cells12050766

**Published:** 2023-02-28

**Authors:** Ozioma S. Chioma, Elizabeth Mallott, Binal Shah-Gandhi, ZaDarreyal Wiggins, Madison Langford, Andrew William Lancaster, Alexander Gelbard, Hongmei Wu, Joyce E. Johnson, Lisa Lancaster, Erin M. Wilfong, Leslie J. Crofford, Courtney G. Montgomery, Luc Van Kaer, Seth Bordenstein, Dawn C. Newcomb, Wonder Puryear Drake

**Affiliations:** 1Departments of Medicine, Vanderbilt University School of Medicine, Nashville, TN 37232, USA; 2Department of Biology, Washington University in St. Louis, St. Louis, MO 63130, USA; 3Otolaryngology-Head and Neck Surgery, Vanderbilt University School of Medicine, Nashville, TN 37232, USA; 4Pathology, Microbiology, and Immunology, Vanderbilt University School of Medicine, Nashville, TN 37232, USA; 5Genes and Human Disease Research Program, Oklahoma Medical Research Foundation, Oklahoma City, OK 73104, USA; 6Department of Biology and Entomology, Pennsylvania State University, College Station, PA 16801, USA

**Keywords:** estrogen, gut microbiome, lung fibrosis, sarcoidosis, Th17 cells

## Abstract

Although profibrotic cytokines, such as IL-17A and TGF-β1, have been implicated in the pathogenesis of interstitial lung disease (ILD), the interactions between gut dysbiosis, gonadotrophic hormones and molecular mediators of profibrotic cytokine expression, such as the phosphorylation of STAT3, have not been defined. Here, through chromatin immunoprecipitation sequencing (ChIP-seq) analysis of primary human CD4+ T cells, we show that regions within the STAT3 locus are significantly enriched for binding by the transcription factor estrogen receptor alpha (ERa). Using the murine model of bleomycin-induced pulmonary fibrosis, we found significantly increased regulatory T cells compared to Th17 cells in the female lung. The genetic absence of ESR1 or ovariectomy in mice significantly increased pSTAT3 and IL-17A expression in pulmonary CD4+ T cells, which was reduced after the repletion of female hormones. Remarkably, there was no significant reduction in lung fibrosis under either condition, suggesting that factors outside of ovarian hormones also contribute. An assessment of lung fibrosis among menstruating females in different rearing environments revealed that environments favoring gut dysbiosis augment fibrosis. Furthermore, hormone repletion following ovariectomy further augmented lung fibrosis, suggesting pathologic interactions between gonadal hormones and gut microbiota in relation to lung fibrosis severity. An analysis of female sarcoidosis patients revealed a significant reduction in pSTAT3 and IL-17A levels and a concomitant increase in TGF-β1 levels in CD4+ T cells compared to male sarcoidosis patients. These studies reveal that estrogen is profibrotic in females and that gut dysbiosis in menstruating females augments lung fibrosis severity, supporting a critical interaction between gonadal hormones and gut flora in lung fibrosis pathogenesis.

## 1. Introduction

An ever-growing synergy of human and animal investigations supports the important role of sex hormone regulation relating to immunity in the pathophysiology of chronic lung diseases [1,2]. IL-17 signaling has been implicated in numerous chronic lung diseases, such as idiopathic pulmonary fibrosis (IPF), lung cancer and pulmonary sarcoidosis [1,3,4,5]. Moreover, striking clinical disparities according to sex are observed in Th17 cell-mediated diseases. For example, although the incidence of IPF is higher in men, being of the female sex is predictive of better IPF clinical outcomes [6,7,8]. Among patients with pulmonary arterial hypertension, female patients have better survival than males [9,10,11]. These observations support the urgent need to identify relevant sex-specific mechanisms in chronic pulmonary inflammation.

Independent reports demonstrate that profibrotic signaling pathways converge on STAT3, an important molecular checkpoint for tissue fibrosis [12,13]. Immune cells, including CD4+ T cells, produce IL-6, which enhances collagen production through the induction of JAK/STAT3/IL-17A or JAK/ERK/TGF-β1 signaling in local and systemic environments [14,15,16,17]. Distinctions in clinical outcomes by sex support an investigation of the interplay of female gonadotrophic hormones with the STAT3-dependent induction of profibrotic cytokine expression. The interactions of the alpha subunit of the estrogen receptor (ERα) and STAT3 protein, both transcription factors, have been reported in breast cancers of epithelial origin, noting enhanced epithelial–mesenchymal transition (EMT) as well as augmented tumor metastasis [18]. However, the immunologic consequences of ERα binding to the *STAT3* gene in CD4+ T cells of patients with lung fibrosis remain unexplored.

The observed disparate clinical outcomes in chronic lung diseases by sex support the investigation of the impact of gonadotrophic hormones on STAT3 signaling, specifically in the context of the profibrotic cytokines, IL-17A and TGF-β1. Here, we report that human females experiencing a loss of lung function due to progressive fibrosis, as well as female murine models of bleomycin-induced lung fibrosis, demonstrate increased T regulatory cells with TGF-β1 expression (immunosuppressive) in the fibrotic lung microenvironment. Lower estrogen states, such as those found in males and ovariectomized female mice, reveal increased IL-17A expression due to elevated percentages of pulmonary Th17 cells (pro-inflammatory). Moreover, the investigation of this estrogen–adaptive immunity interplay in distinct environments reveals that low gut microbial diversity further increases estrogen-induced lung fibrosis. These data demonstrate a distinct sex-specific role for STAT3 signaling in CD4+ T cells, thus paving the way for developing personalized (e.g., sex-based) immunotherapeutic strategies for chronic lung inflammation.

## 2. Materials and Methods

### 2.1. Human Study Approval

To participate in this study, all of the human subjects signed a written informed consent form, and the patients were enrolled at Vanderbilt University Medical Center. All of the human studies were approved by the appropriate institutional review board (VUMC 040187).

### 2.2. Study Population

For inclusion in this study, the clinical and radiographic criteria used to define sarcoidosis were applied [19]. IPF subjects were defined according to recent American Thoracic Society (ATS) guidelines [20], and systemic sclerosis patients were defined according to the 2013 American College of Rheumatology criteria [21]. Clinical lung progression was defined as previously described [22]. Pulmonary function testing was performed as clinically indicated. FVC decline was defined as a relative reduction of ≥10% in the percent of predicted FVC. There were four human cohorts in this study: 25 healthy controls (7 males and 18 females), 31 sarcoidosis patients (11 males and 20 females), idiopathic pulmonary fibrosis (IPF) patients (36 males and 9 females), and scleroderma patients (5 males and 6 females). Information related to the demographics of the study subjects is provided in Table 1.

### 2.3. Peripheral Blood Mononuclear Cells Isolation and Storage

The Ficoll–Hypaque density gradient centrifugation method was used to isolate peripheral blood mononuclear cells (PBMCs) from the whole blood of all four human cohorts in this study: healthy controls, sarcoidosis, IPF, and scleroderma patients, as previously described [23,24]. The PBMCs were then stored in fetal bovine serum containing 10% dimethyl sulfoxide (DMSO) at a concentration of 10 × 10^6^ cells/mL in a −80 °C freezer before being transferred to liquid nitrogen for prolonged storage or before use. 

### 2.4. Chromatin Immunoprecipitation Sequencing (ChIP-Seq) Library Preparation

Primary CD4+ T cells were negatively selected using immunomagnetic bead separation (STEMCELL, EasySep #17951). Approximately 1 to 2.5 million total T cells were obtained from 5 to 10 million PBMCs. The T cells were first incubated with 2 mM disuccinimidyl glutarate for 35 min at room temperature; then, formaldehyde was added to a final concentration of 1%, and the cells were incubated for another 10 min at room temperature [25]. The nuclei were isolated using the Covaris truChIP Chromatin Shearing Kit and fragmented by sonication. Immunoprecipitation was performed using an anti-ERα antibody (Cell Signaling #8644) and protein A+G magnetic beads. The chromatins were de-crosslinked and purified using AMPure XP beads. ChIP-seq libraries were prepared according to Illumina protocols and were sequenced using 75 bp paired-end sequencing on an Illumina NextSeq, producing an average of 135,924,844 reads per library.

### 2.5. Sequencing Alignment and Peak Calling

The ChIP-seq reads were examined for technical artifacts using FastQC. No aberrant technical behavior was identified. The reads were trimmed for adapter sequences and decontaminated for sequencing artifacts by using bbduk. The trimming options were set to ktrim = right trimming, mink = 11, hdist = 1, qin = 33, tpe and tbo options enabled. BBDuk’s list of Illumina sequencing adapters was used to perform adapter trimming. Decontamination was performed against phiX adapters and bbduk’s database of sequencing artifacts. The decontaminated reads were aligned to version GRCh38 of the human reference genome using BWA-mem [26], with the following options: -L 100 -k 8 -O 5. Following the alignment, the peaks were called with respect to the input chromatin library using MACS2 [27], with the following options: -nomodel –shift -100 –extsize 200 g hs -q 0.05 -f BAMPE –keep-dup all –broad.

### 2.6. Murine Model of Pulmonary Fibrosis

All of the murine procedures were performed according to the protocol approved by the Institutional Animal Care and Use Committee at Vanderbilt University Medical Center (protocol #M1700043). For the murine model of bleomycin-induced pulmonary fibrosis, 5- to 8-week-old mice weighing approximately 17–22 g were used. The mice were anesthetized with an intraperitoneal injection of 80 μL of 20 mg/mL Ketamine/1.8 mg/mL Xylazine solution. Then, 75 μL containing 0.04 units of bleomycin (Novaplus Lake Forest IL) in saline or an equal volume of saline (0.9% sodium chloride) (Hospira Inc., Lake Forest IL), used as a control, was administrated intranasally to wild-type or ESR-1-/- mice, as previously described [28]. The lungs were harvested for histology, flow cytometry, or single-cell isolation, as previously described [16]. The mouse strains used are described in Appendix A.

### 2.7. Ashcroft Scoring

The degree of fibrosis in the murine lung tissue was assessed using Ashcroft scoring, as previously described [29].

### 2.8. Sircol Assay

The collagen content was determined using a Sircol Collagen Assay kit (Biocolor, Newtown Abbey, UK), as previously described [30].

### 2.9. Flow Cytometry

Both murine and human flow cytometry experiments were conducted with an LSR-II flow cytometer (BD Biosciences, Franklin Lakes, NJ, USA), and the information related to all the antibodies used in this study is listed in Appendix A. Live cells were gated based on the forward and side scatter properties, and the surface staining of cells was performed as previously described [31]. Th17 cells were identified by flow cytometry using key transcriptional factors, such as STAT3, as previously described [32]. The cells were gated on singlets, live CD3+ and CD4+ cells. Data analysis was performed using FlowJo software (Tree Star, Ashland, OR, USA). A minimum of 50,000 events were acquired per sample.

### 2.10. In Vivo Implantation of Hormone Pellets to Ovariectomized Mice

Ovariectomy or sham surgeries were conducted at three weeks of age by the Jackson Laboratory, and the experiments were carried out when the ovariectomized or sham-operated mice were 6 weeks old. At 6 weeks of age, 60-day slow-release pellets (Innovate Research of America, Sarasota, FL, USA) containing 17β- estradiol 0.1 mg (E2), progesterone 25 mg (P4) or a combination of 17β-E2 (0.1 mg) and P4 (25 mg) were surgically placed subcutaneously into ovariectomized C57BL/6J mice, as previously described [33]. As a control, 25.1 mg of vehicle pellets (Innovative Research of America) was surgically placed into the sham-operated females or ovariectomized female mice. Three weeks (21 days) after the pellets were implanted, the mice were challenged with intranasal bleomycin (0.04 Units) and sacrificed 14 days later, as previously described [28]. Studies involving large and independent experimental cohorts of mice were performed at least twice.

### 2.11. Metagenomic Sequencing and Analysis of Gut Microbiota

Fecal pellets were collected from female mice at Day 14 in each housing cohort, and genomic DNA (gDNA) was extracted with the Qiagen DNAeasy extraction kit (Qiagen, Valencia, CA, USA), according to the manufacturer’s instructions. The gDNA concentration and quality were confirmed using the Bioanalyzer 2100 system (Agilent, Santa Clara, CA, USA). The metagenomic sequencing and analysis of fecal pellets was conducted as previously described [34]. The sequences of gut microbiota have been deposited into BioProject ID PRJNA899808. Wilcoxon Rank Sum tests in R were used to examine differences in Shannon diversity and evenness between the ABSL-1 and ABSL-2 environments. The code for all of the analyses can be found at http://github.com/emallott/PulmonaryFibrosisMicrobiota (accessed on 19 January 2023) [34].

### 2.12. Statistics

When comparing different experimental groups, we used an unpaired two-tailed Student’s *t*-test. Multiple-group comparisons were performed using a one-way analysis of variance (ANOVA) with Tukey’s post hoc test. Statistical analysis for all figures was carried out using Prism version 7.02 (GraphPad Software, San Diego, CA, USA). For a result to be considered statistically significant, a *p*-value of less than 0.05 was used.

## 3. Results

### 3.1. The Nuclear Transcription Factor, Estrogen Receptor Alpha Subunit, Interacts with the STAT3 Gene Locus in CD4+ T Cells

The estrogen receptor alpha subunit (ERα) is not only a receptor but also serves as a transcription factor. To identify factors that may modulate *STAT3* expression during lung fibrosis, we interrogated ChIP-seq datasets in the ENCODE 3 repository [35]. In five human cell lines, including cancers and EBV-transformed B lymphocytes, a significant enrichment of ERα binding was demonstrated within the *STAT3* locus. Representative tracks among the technical replicates for each cell line were visualized in the WashU Epigenome Browser [36] (Figure 1A). The numbers of starting reads, decontaminated reads, alignment successes, and enriched peaks are given in Appendix A. These findings in the Chip-seq datasets confirmed previous reports indicating that ERα, which is encoded by the *ESR1* gene, and STAT3 are important in breast and ovarian cancer [18,37], supporting the hypothesis that the *STAT3* gene locus is a frequent target of ERα activity in various cell types. 

The targeting of ERα to the *STAT3* gene locus in T cells has not been previously described. To determine whether ERα interacts with the *STAT3* locus in CD4+ T cells through DNA binding activity, we performed genome-wide ChIP-seq for Erα-bound regions. Primary CD4+ T cells were derived from the PBMCs of six healthy individuals with varying demographics (Appendix A). Of the six ChIP libraries (four females and two males), sample p1035928-8, which corresponds to a female, identified an over sixfold greater number of ERα-enriched regions relative to any other sample. We used the GREAT algorithm to perform ontology-based functional enrichment analyses on that sample. ERα-enriched sites were statistically significantly enriched in genes related to T-cell function and development (Appendix A), suggesting that the peaks obtained from this ChIP capture are specific to CD4+ T-cell function and are not randomly organized across the genome.

Finally, we examined the *STAT3* locus in detail. We found that sample p1035928-8 contains six ERα-binding regions within or proximal to the *STAT3* genomic locus, including two in its promoter region (Figure 1B). Overlaying chromatin accessibility data from the ENCODE project [35], we noted that each of these regions exhibits DNase hypersensitivity in at least one ENCODE cell line. Three of these regions also displayed evidence of estrogen-related receptor alpha (ESRRA) binding in other cell lines (K562, GM12878). Taken together, these results demonstrate that ERα binds the *STAT3* locus in CD4+ T cells, specifically at known regions of chromatin accessibility shared with various cell types.

### 3.2. Loss of the ESR-1 Subunit Represses IL-6 Expression but Augments pSTAT3 and IL-17A Expression in CD4+ T Cells

Because transcription factor ESR-1 (alpha subunit of ESR) was identified as binding to the STAT3 gene in CD4+ T cells, we investigated the role of the ERα subunit in profibrotic cytokine expression using a murine model of bleomycin-induced lung fibrosis in WT and ESR-1 knockout (ESR-1-/-) mice. Both murine cohorts were challenged intranasally with bleomycin and harvested on day 14. ESR-1-/- mice contain supernormal estrogen levels in their serum due to the loss of the ESR-1 signaling-mediated negative feedback loop [38]. We observed that female ESR-1-/- mice lost significantly less weight and had the same mortality compared to their WT counterparts (Figure 2A,B). Male ESR-1-/- mice also demonstrated reduced weight loss but had significantly increased survival compared to WT males (Appendix A). We used flow cytometry to examine profibrotic cytokine expression in pulmonary CD4+ T cells of the murine cohorts. The levels of IL-6 and IL-23R, key mediators of Th17 cell differentiation, were significantly reduced in the lung CD4+ T cells of female ESR-1-/- mice compared to their WT counterparts (Figure 2C,D). Remarkably, the levels of pSTAT3 and IL-17A were increased in ESR-1-/- compared with WT mice (Figure 2E,F). These data demonstrate that ESR-1 has a key role in the induction of IL-6 and IL-23R expression in CD4+ T cells, as well as the repression of pSTAT3 and IL-17A expression in CD4+ T cells during the pulmonary fibrosis of females.

### 3.3. Loss of Gonadotrophic Hormones through Ovariectomy Reduces IL-6 Production and Augments pSTAT3 and IL-17A Expression from CD4+ T Cells

To further delineate the contribution of female gonadotrophic hormonal signaling to the progression of proinflammatory cytokine expression in the lung, we used female C57BL/6J mice that were ovariectomized or sham-operated at three weeks of age. Slow-release pellets containing either 17β-estradiol (17β-E2, 0.1 mg), progesterone (P4, 25 mg), the combination of 17β-E2 (0.1 mg) and P4 (25 mg) or a vehicle (25.1 mg) were subcutaneously implanted into adult ovariectomized female C57BL/6J mice at six weeks of age. At nine weeks of age, all groups were challenged with bleomycin, and the lungs were harvested 14 days later. There was no significant difference in weight loss or survival across the hormone treatment groups compared to the ovariectomized mice implanted with placebo pellets (Figure 3A,B). 

We performed flow cytometric analysis of single-cell lung suspensions (SCLS) to assess alterations of CD4+ T cell populations. TGF-β and IL-17A are profibrotic cytokines that are expressed by regulatory T cells and Th17 cells, respectively. We began by comparing regulatory T and Th17 cell populations in sham-operated, menstruating female mice. We noted a significantly higher population of regulatory T cells compared to Th17 cells in the sham-operated mice (Figure 3C). We then assessed for IL-17A cytokine expression in response to the loss of female hormones. Ovariectomized mice displayed decreased CD4+IL-6+ T cells compared to the sham-operated mice; supplementation with both 17β-E2 and P4 in ovariectomized mice normalized IL-6 expression. Neither hormone individually restored IL-6 expression by CD4+T cells to the same levels as the sham-operated mice (Figure 3D). The same trends held for the IL-6 co-receptor GP130 (Figure 3E). Remarkably, and akin to our observation in ESR-1-/- mice, the levels of pSTAT3 were increased in the CD4+ T cells of ovariectomized mice compared to sham-operated animals, again returning to sham levels in ovariectomized mice by the addition of female hormones (Figure 3F). In accordance with an increase in pSTAT3, we also observed heightened CD4+IL-17A+ T cells in ovariectomized mice compared to sham-operated animals. The addition of 17β-E2, P4 or both to ovariectomized mice decreased IL-17A expression compared to the placebo (Figure 3G). A representative FACS plot is provided (Figure 3H). Overall, these findings reveal that female hormones repress inflammatory profibrotic cytokine expression by inhibiting pSTAT3 signaling and IL-17A expression in murine pulmonary CD4+ T cells following bleomycin administration.

### 3.4. Lung Quantification following the Loss of ESR-1 or Ovariectomy Reveals Reduced Collagen Content

To determine the physiologic significance of estrogen signaling for profibrotic cytokine expression, we performed histologic analysis and collagen quantification of the lung using the Sircol assay. Analysis of lung histology using trichrome staining noted significantly less fibrosis in ovariectomized mice without hormone replacement compared to the sham-operated mice or ovariectomized mice given dual estrogen (17β-E2)/progesterone (P4) hormone pellets (Figure 4A). Ashcroft scoring (Appendix A) and the quantification of collagen content (Figure 4B) revealed a nonsignificant decrease in collagen levels in ovariectomized mice compared to mice that underwent sham ovariectomy surgeries. The replacement of female hormones with a combination of estrogen and progesterone pellets increased fibrosis compared to the ovariectomized placebo group (Figure 4B). Similarly, a nonsignificant decrease in pulmonary collagen content was observed in ESR-1-/- mice compared to wild-type mice. The observation of a nonsignificant decline in the pulmonary lung content following the loss of estrogen signaling suggests that additional factors contribute to pathogenesis. We recently reported that the gut microbiota play an important role in lung fibrosis severity. ABSL-1 housing conditions favor gut microbiota diversity, whereas ABSL-2 conditions favor reduced gut microbiota diversity [34]. Using linear discriminant analysis (LDA) to examine species-level differences in the gut microbiota, 10 taxa were overrepresented in ABSL-1 mice, and five taxa were overrepresented in ABSL-2 mice. The overrepresented taxa in ABSL-2 mice included Lachnospiraceae bacterium A2, Lachnospiraceae bacterium 28–4, Firmicutes bacterium ASF500 and Romboutsia ilealis [34]. A higher relative abundance of Firmicutes in the lung microbiota of bleomycin-treated mice with fibrosis has been reported [39]. The species overrepresented in ABSL-1 mice included *Staphylococcus nepalensis, Dubosiella newyorkensis, Acetatifactor muris, Lactobacillus animalis, Lactobacillus murinus* and *Acutalibacter muris* [34].

No distinctions in the lung microbiota are present in these mice regarding the housing condition. Specifically, rearing environments that favor low gut microbiota diversity, such as ABSL-2 housing conditions, induce severe lung disease compared to ABSL-1 conditions. To confirm if gut microbiota impact female ILD severity, we began by assessing the lung collagen content in wild-type female mice who received intranasal bleomycin while housed in different environments: germ-free, ABSL-1 or ABSL-2 conditions. We noted significant distinctions in lung collagen content among wild-type females according to the rearing environment, with ABSL-2 female mice demonstrating the most severe disease compared to germ-free or ABSL-1 mice (Figure 4D). To determine the impact of estrogen signaling and gut microbiota on lung fibrosis severity, we assessed the lung collagen content among ovariectomized mice, as well as those ovariectomized with estrogen replacement, while housed under either ABSL-1 or ABSL-2 conditions. Remarkably, we noted that ovariectomized mice housed under ABSL-1 or ABSL-2 conditions did not demonstrate a change in the collagen content (Figure 4E). Equally noteworthy was the observation that a significant increase in lung fibrosis was noted among ovariectomized mice who received estrogen replacement and were housed in ABSL-2 conditions compared to those housed in ABSL-1 conditions. These findings reveal a synergistic relationship between estrogen signaling and gut dysbiosis regarding lung fibrosis severity (Figure 4E).

### 3.5. Female Gut Microbiota Demonstrate Significantly Less Diversity in ABSL-2 Housing Conditions

To investigate the hypothesis that the gut microbiota is an important contributor to the differences in fibrosis severity between female mice housed under ABSL-1 and ABSL-2 conditions, we performed metagenomic analysis on fecal pellets from female mice in each housing cohort. We did not detect microorganisms in the stool of female germ-free mice by sequencing and culture, as expected. Shannon alpha diversity, a measure of species richness and evenness, was considerably higher in female ABSL-1 mice compared with female ABSL-2 mice using a Wilcoxon rank sum test (Figure 5A). Additionally, Pielou’s evenness was higher in ABSL-1 compared with ABSL-2 female mice, but species richness did not differ significantly (Shannon diversity: Wilcoxon, W = 108, *p* = 0.015; Pielou’s evenness: Wilcoxon, W = 106, *p* = 0.021; Species richness: Wilcoxon, W = 80.5, *p* = 0.450). The female mice housed under ABSL-1 and ABSL-2 conditions differed significantly in their gut microbiome composition using Jaccard but not Bray–Curtis dissimilarities (PERMANOVA, Bray–Curtis: F1,22 = 2.392, R2 = 0.098, *p* = 0.079; Jaccard: F1,22 = 8.369, R2 = 0.276, *p* < 0.001). A similar investigation in male mice revealed that the ABSL-1 and ABSL-2 microbiomes were significantly different using both metrics (PERMANOVA, Bray–Curtis: F1,24 = 4.728, R2 = 0.165, *p* = 0.004; Jaccard: F1,24 = 6.519, R2 = 0.214, *p* < 0.001) (Figure 4). Alpha diversity did not differ significantly between floors for male individuals (all *p* > 0.05).

A comparison of female and male gut microbiota diversity according to the housing conditions reveals significantly greater gut diversity among females compared to males under ABSL-1 housing conditions (Figure 5B), whereas only greater species richness was noted among females under ABSL-2 housing conditions (Figure 5C). Beta diversity differences between ABSL-1 and ABSL-2 microbiota compositions also differed significantly when an analysis was conducted using both the Bray–Curtis dissimilarity metric index (Figure 5D) and the Jaccard index (Figure 5E), which account for the presence/absence of taxa and taxon abundance variation, respectively (PERMANOVA, ABSL-1 mice: Bray–Curtis: F_1,17_ = 4.424, R^2^ = 0.206, *p* = 0.014; Jaccard: F_1,17_ = 2.408, R^2^ = 0.124, *p* = 0.053; ABSL-2 mice: Bray–Curtis: F_1,29_ = 1.952, R^2^ = 0.063, *p* = 0.160; Jaccard: F_1,29_ = 7.944, R^2^ = 0.215, *p* < 0.001). These findings support the hypothesis that the female gut microbiome changes according to the rearing environment.

### 3.6. Patients with Progressive Fibrotic Lung Disease Display Sex-Specific Profibrotic Cytokine Profiles

Because of the role of female gonadotrophic hormones in reducing the CD4+ T cell-mediated proinflammatory and profibrotic environment in mouse models of lung fibrosis, we probed samples from human patients with fibrotic lung diseases for sex-associated differences. Consistent with the murine model of lung fibrosis, we observed higher levels of STAT3 mRNA and pSTAT3 protein in CD4+ T cells from the male compared to the female sarcoidosis patients (Figure 6A,B). We noted similarly increased mRNA and protein expression of the master transcription factor regulating IL-17A production, RORC, in CD4+ T cells from the male compared to the female sarcoidosis patients (Figure 6C,D). Additionally, among sarcoidosis patients experiencing disease progression, females expressed significantly higher IL-6 levels in their CD4+ T cells compared to males (Figure 6E).

We also assessed IL-17A and TGF-β1 production by sex, as CD4+ T cells promote pulmonary fibrosis through the STAT3-medicated production of IL-17A and TGF-β1 [16]. We observed higher IL-17A mRNA and protein expression in CD4+ T cells from male compared to female sarcoidosis patients (Figure 6F,G). CD4+ T cells from female sarcoidosis patients expressed significantly higher free TGF-b1 than males and the healthy female controls (Figure 6H). There were no distinctions in the TGF-b1 precursor protein, latency-associated peptide-TGF-β, among males compared to females (Figure 6I). These findings demonstrate the differential immune modulation of STAT3 signaling pathways in human CD4+ T cells of males (increased) and females (reduced) with fibrotic lung disease. Consequently, CD4+ T cells from males exhibit higher proinflammatory cytokine expression due to enhanced IL-17A production, whereas CD4+ T cells from females exhibit increased immunosuppressive cytokines due to greater TGF-β1 expression.

We assessed for a possible contribution of female hormones to other fibrotic diseases, including IPF and Systemic Sclerosis (SSc), by quantifying the serum 17β-E2 levels in age-matched patients and healthy controls. Serum 17β-E2 was greater in male SSc and IPF patients compared to age-matched male healthy controls (Figure 6J). These findings demonstrate the positive interplay of female gonadotrophic hormones in male- and female-predominant fibrotic lung diseases.

## 4. Discussion

This original report reveals the “ying–yang” effects of estrogen-induced lung fibrosis in female interstitial lung disease. Estrogen clearly augments the development of lung fibrosis (Figure 4); yet, the binding of ERα to the STAT3 promoter shifts profibrotic cytokine expression away from proinflammatory phenotypes mediated by IL-17A to immunosuppressive phenotypes mediated by TGF-β1 (Figure 2 and Figure 3). Human cytokine expression confirmed reduced pSTAT3 expression in females, leading to increased TGF-β1 production, whereas males display higher IL-17A levels.

The beneficial effects of estrogen were apparent. Although ESR-1-/- mice and surgical ovariectomy confirm estrogen’s profibrotic capacity in lung fibrosis, it is worth noting that Th17 cell differentiation is reduced due to the transcription factor ERα‘s ability in relation to the STAT3 promoter (Figure 1, Figure 2 and Figure 3). The loss of STAT3 signaling has been shown to shift the IL-6-JAK2-STAT3 induction of IL-17A to sustained IL-6-ERK-TGF-β1 expression in local and systemic CD4+ T cells [15,16]. This is the most likely explanation for the increased regulatory T cells noted in females and the increased STAT3 signaling and IL-17A production following ovariectomy (Figure 3 and Figure 6). Both ovariectomized and ESR-1-/- mice revealed significantly lower IL-6 and GP130 levels than sham-treated animals but increased pSTAT3 and IL-17A levels in CD4+ T cells (Figure 3). Higher estrogen states augment IL-6 production, but instead of inducing a proinflammatory state supported by increased CD4+ IL-17A levels, estrogen concomitantly inhibits STAT3 signaling. These immune alterations are likely relevant to other IL-17A-mediated diseases in the postmenopausal state, such as myocardial infarctions and osteoporosis [40,41]. Enhanced TGF-β1 expression protects against osteoporosis [42].

The pathologic effects of estrogen were also determined. A prior study noted increased ESR-1 expression in human IPF lung samples and that the chemical inhibition of ESR-1 results in reductions in bleomycin-induced pulmonary fibrosis in male mice [43]. The genetic and surgical ablation of estrogen-dependent signaling resulted in reductions in the pulmonary collagen content, which confirms the profibrotic nature of estrogen in female-predominant ILD (Figure 4). Remarkably, the observed reductions were not statistically significant, suggesting that other factors contribute to lung fibrosis severity in females. The induction of lung fibrosis in females under distinct housing conditions unveiled the role of the gut microbiome in lung fibrosis severity. Wild-type female mice treated with intranasal bleomycin demonstrate the greatest lung severity under ABSL-2 conditions and minimal fibrosis under germ-free conditions, thus confirming the important contribution of gut flora to female lung fibrosis (Figure 4D). Conditions that favor the loss of female gut microbial diversity, such as ABSL-2 housing conditions, lead to greater lung fibrosis compared to ABSL-1 conditions (Figure 4 and Figure 5). Equally noteworthy is the observation that fibrosis is synergistic between estrogen signaling and gut dysbiosis, suggesting that the profibrotic nature of estrogen is heavily influenced by gut microbiota and that the capacity of gut microbiota to induce fibrosis is influenced by the host hormone status. A growing body of literature supports crucial interactions between gut microbiota and estrogens [44,45]. The conjugation of glucuronic acid (GlcA) to a compound, such as estrogen, marks it for elimination via the GI or urinary tract. β-glucuronidase, an enzyme that deconjugates estrogen, mediates estrogen release into the serum in its active form [46,47]. Gut microbiota can inhibit or induce β-glucuronidase activity. In addition, it was previously noted that ABSL-2 stool contains reduced lactobacilli within the microbial community. Lactobacillus spp, which were elevated in ABSL-1 stool, can reduce fecal β-glucuronidase activity [45]; future studies that assess the capacity of lactobacilli to enhance urinary estrogen excretion and lower its serum levels are needed. Future studies defining the mechanisms by which ABSL-2 gut flora augment the estrogen induction of lung fibrosis are also warranted. Considering of the hormone status of the host, as well as defining the gut microbiome, is necessary to explain the clinical observations in females with ILD. TGF-β1 is the master regulator of fibrosis. Figure 6 demonstrates that TGF-β1 is most predominant in female sarcoidosis patients. In Figure 4, we see that gut dysbiosis augments lung fibrosis. When IL-6 induction occurs, downstream signaling can lead to either IL-17A or TGF-B1 expression. IL-17A expression leads to pulmonary inflammation. Estrogen signaling provides protection against proinflammatory fibrosis due to the capacity of the ERα to bind to the STAT3 promoter (Figure 1). This reduction in lung inflammation improves the prognosis. In menopausal females, the gut microbiome continues to drive lung fibrosis, but due to the reduced estrogen state, there is no inhibition of STAT3 expression and Th17 cell development. Lung fibrosis can now be mediated by IL-17A, which likely explains the increased symptoms after menopause.

There are some limitations that should be noted. This investigation focused on female ILD; investigations of the role of testosterone in lung fibrosis are needed. There are also reports indicating that estrogen drives Th17 cell differentiation in chronic lung diseases, such as asthma [48,49]. Concomitant immune-gut microbiome investigations of asthma models with ILD models are warranted, including an inquiry into the interplay of gonadal hormones. Additionally, asthma pathogenesis is very distinct from ILD, which may also impact T cell differentiation. Another consideration is that the gut microbiome is influenced by diet. The mice in the murine model had the same diet; future studies assessing the impact of food consumption on the gut microbial community, metabolomic syndromes and inflammation are warranted [50,51,52]. An investigation into the impact of gut dysbiosis on estrogen signaling or of estrogen signaling on gut microbial communities is warranted. Finally, we observed Th17 cell populations increasing following the gavage of ABSL-2 stool into germ-free mice compared to the gavaging of ABSL-1 stool. Future analysis definitively identifying the microorganism(s) responsible for Th17 cell differentiation is warranted, followed by an assessment of their presence in the stool of murine asthma models, as well as asthmatic patients and ILD patients.

## 5. Conclusions

Taken together, this investigation demonstrates that female gonadotrophic hormones are profibrotic yet, through the ERα binding of the *STAT3* locus, reduce the inflammation induced by IL-17A expression in CD4+ T cells. The consequent reduction in inflammation is a likely contributor to the mortality benefit observed in premenopausal females with ILD. This study introduces another key contributor to lung fibrosis severity: gut dysbiosis. The synergistic impact of gut dysbiosis and estrogen on lung fibrosis supports a multi-pronged approach to the treatment of female-predominant lung fibrosis (Figure 7).

## Figures and Tables

**Figure 1 cells-12-00766-f001:**
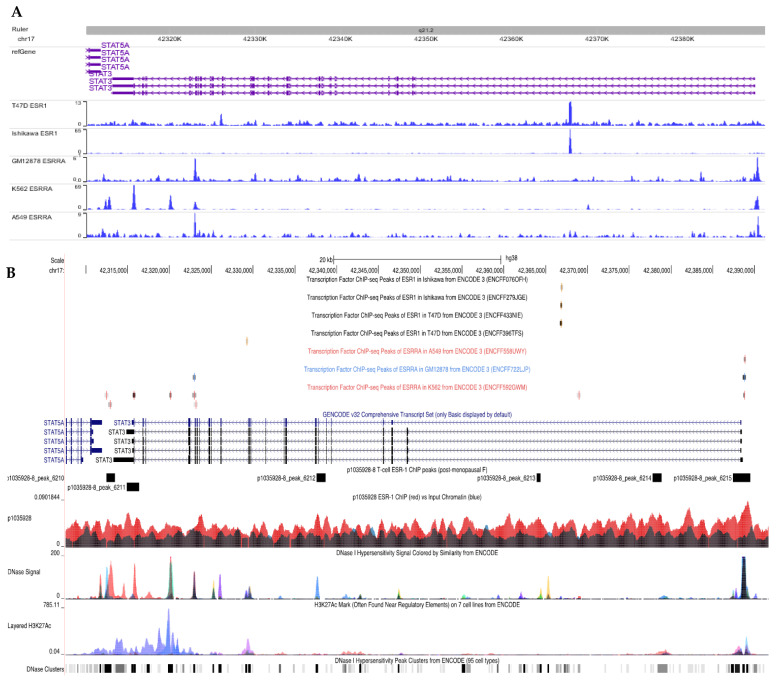
**Genomic visualization of ESR1 or ESRRA binding activity at the STAT3 gene locus.** (**A**) Gene annotations indicate the position of STAT3, its exons (thick bars) and introns (connecting arrows showing the direction of translation). Blue tracks indicate the enrichment of the ChIP-seq signal for ESR1 or ESRRA binding over the background input for five cell lines. Sharp, prominent peaks provide evidence for the ability of these transcription factors to bind STAT3, demonstrating their interaction in these cell types. (**B**) Track visualization of ERα ChIP-seq for the STAT3 locus. Primary CD4+ T-cell ERα ChIP enrichment signal (red) is depicted against the input chromatin signal (black). Rectangular regions above the ChIP track indicate regions significantly enriched with ERα binding. Top: Additional tracks indicate positions of ESR1 or ESRRA binding activity in ChIP-seq data from the ENCODE project. Note that ERα binding events are not typically shared from experiment to experiment, even when cell lines are identical. Bottom: DNase and H3K27ac signal from the ENCODE project, indicating regions of strong enhancer activity in general cell lines. ERα-binding regions in T-cells occur in areas with known chromatin accessibility.

**Figure 2 cells-12-00766-f002:**
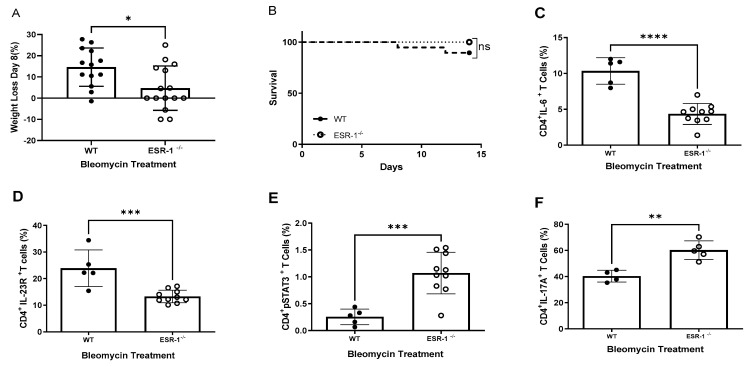
**Loss of estrogen receptor alpha subunit (ESR-1) improves survival and ameliorates fibrosis in female mice**. WT and ESR-/- mice were treated with bleomycin and monitored for 14 days. (**A**) Body weights of mice at day 8 compared to day 0; (**B**) Murine mortality across 14 days. Kaplan–Meier survival analysis with a log-rank test was used to determine differences between groups. Flow cytometric analysis of T cells from single-cell lung suspensions at day 14 for (**C**) IL-6, (**D**) IL-23R, (**E**) pSTAT3Y705 and (**F**) IL-17A. Comparisons between cohorts were performed using one-way ANOVA with Tukey’s post hoc test. * *p* < 0.05, ** *p* < 0.01, *** *p* < 0.001; **** *p* < 0.0001; ns: no significance. Bars are the mean ± SD; each symbol represents an individual mouse. WT: Wild-type, ESR-1-/- estrogen receptor alpha knockout mice. N = 4–15.

**Figure 3 cells-12-00766-f003:**
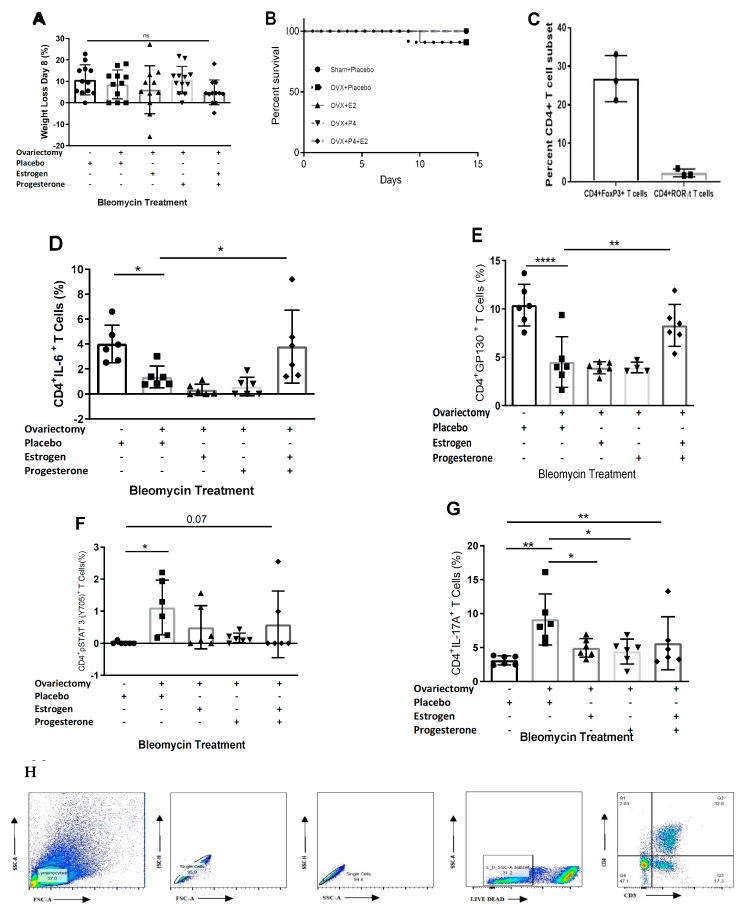
**In vivo administration of female hormones increases profibrotic cytokine expression in ovariectomized mice**. Hormone-containing or placebo pellets were implanted into ovariectomized C57BL/6 female mice for 21 days, followed by bleomycin administration and monitoring for an additional 14 days. (**A**) Body weights of mice at day 8 after bleomycin administration; (**B**) Murine mortality across 14 days; (**C**) Flow cytometric analysis of T cells from single-cell lung suspensions for (**D**) IL-6, (**E**) GP130, (**F**) pSTAT3Y705 and (**G**) IL-17A. (**H**) Representative FACS plots illustrating CD4+ percentage in bleomycin-treated murine cohorts. Comparisons between cohorts were performed using one-way ANOVA with Tukey’s post hoc test. * *p* < 0.05, ** *p*< 0.01, **** *p* < 0.0001. ns: no significance; RLL: right lower lobe. Bars are the mean ± SD; each dot represents an individual mouse. N = 3–12 per cohort.

**Figure 4 cells-12-00766-f004:**
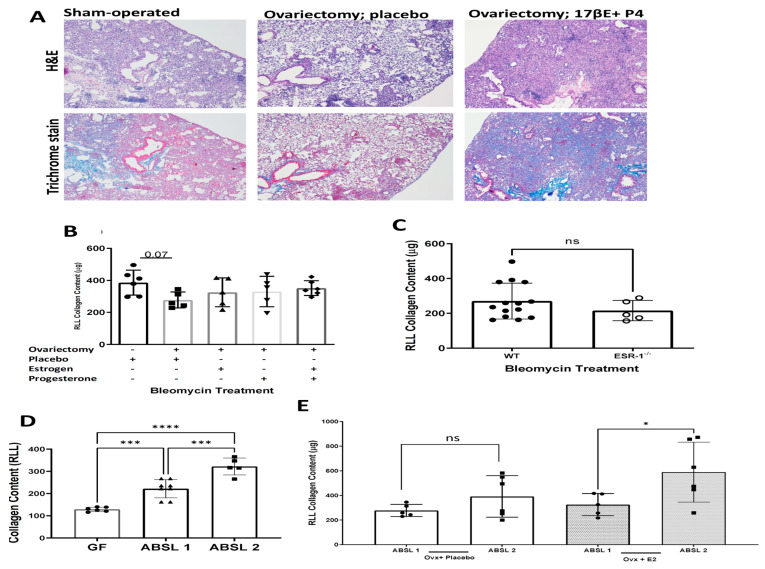
**Quantification of bleomycin-induced pulmonary collagen with various hormone deletion or repletion conditions in distinct housing environments**. (**A**) Representative H&E and trichrome histologic stains of lungs at day 14 under various hormone conditions; (**B**) Pulmonary collagen quantification of the lung under baseline, hormone depletion and hormone repletion conditions. (**C**) Pulmonary collagen quantification of the lung under wild-type (WT) and ESR1 null conditions; (**D**) Pulmonary collagen quantification in menstruating female mice housed in germ-free, ABSL-1 and ABSL-2 environments; (**E**) Pulmonary collagen quantification of ovariectomized and estrogen-repleted mice in ABSL-1 and ABSL-2 environments. Comparisons between cohorts were performed using one-way ANOVA with Tukey’s post hoc test. * *p* < 0.05, *** *p*< 0.001, **** *p* < 0.0001. ns: no significance; RLL: right lower lobe. Bars are the mean ± SD; each dot represents an individual mouse. N = 5–14.

**Figure 5 cells-12-00766-f005:**
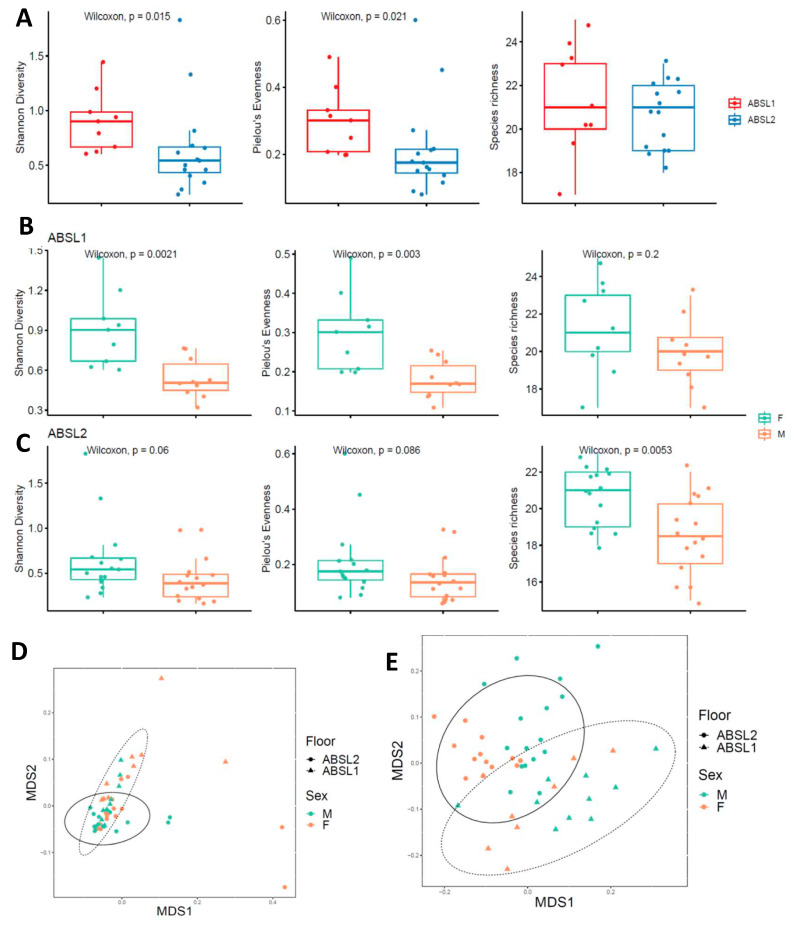
**Murine female gut microbial diversity is modified by housing environment.** (**A**) Shannon diversity index, Pielou’s evenness and species richness scores for female mice housed in ABSL-1 and ABSL-2 facilities following bleomycin inoculation (*N*= 9–14 mice per cohort). The boxes show the median, as well as the 25th and 75th quartiles. The whiskers extend to 1.5*IQR. Each dot represents one mouse. (**B**,**C**) Comparison of female and male Shannon diversity index, Pielou’s evenness and species richness scores for female mice housed in ABSL-1 and ABSL-2 facilities following bleomycin inoculation (n =14–16 mice per cohort). (**D**) Nonlinear multidimensional scaling (MDS) plot showing differences in microbial taxonomic composition based on Jaccard dissimilarities. (**E**) Nonmetric multidimensional scaling plot based on Bray–Curtis and Jaccard dissimilarities showing the gut microbiome by gender of murine communities housed on ABSL1 and ABSL2 floors.

**Figure 6 cells-12-00766-f006:**
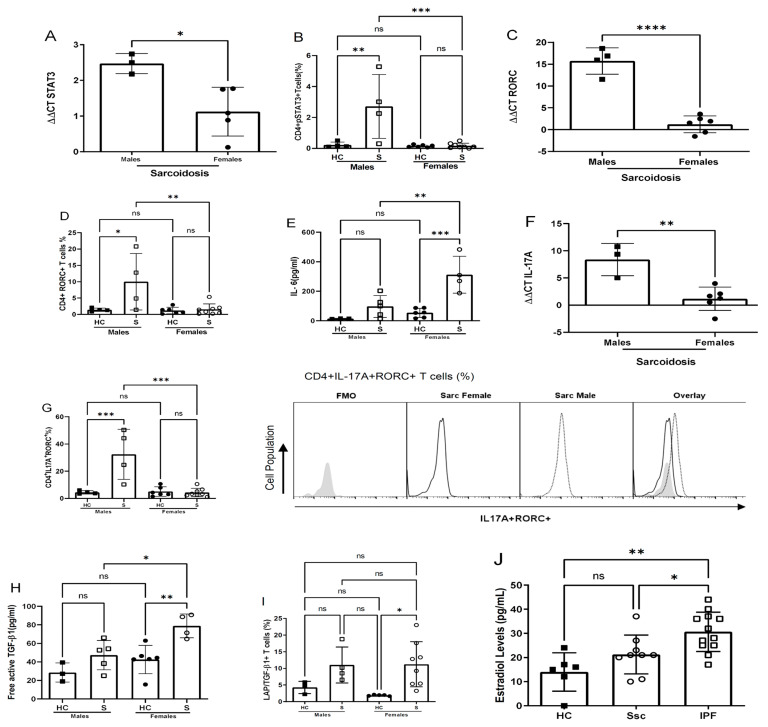
**Male and female sarcoidosis patients display distinct profibrotic cytokine profiles**. Purified CD4+ T cells from the peripheral blood of healthy controls and sarcoidosis patients were anti-CD3- and anti-CD28 TCR-stimulated and cultured for 24 h, followed by real time-PCR for (**A**) STAT3, (**C**) RORC, (**F**) IL-17A; flow cytometry for (**B**) pSTAT3Y705, (**D**) RORC, (**G**) IL-17A, (**I**) LAP/TGF-β1; (**E**) cytokine bead array for IL-6 and (**H**) enzyme-linked immunosorbent assay for free TGF-β1 analysis. (**J**) Estradiol levels in serum of healthy controls, IPF and scleroderma male patients. Comparisons between cohorts were performed using one-way ANOVA with Tukey’s post hoc test. Bars are the mean ± SD; each dot is an individual patient. * *p* < 0.05, ** *p* < 0.01, *** *p* > 0.001, **** *p* < 0.0001. ns: no significance, HC: healthy controls, S: sarcoidosis, Ssc: systemic sclerosis, IFP: idiopathic pulmonary fibrosis.

**Figure 7 cells-12-00766-f007:**
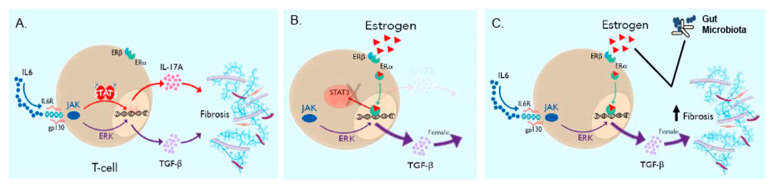
**Graphical abstract of the interaction of gonadal hormones and gut dysbiosis in lung fibrosis**. (**A**) IL-6 induces profibrotic cytokine expression through IL-17A and TGF-β1 expression. IL-17A drives inflammation in fibrotic lung tissue. (**B**) The alpha subunit of the estrogen receptor (ERα) serves as a transcription factor and physically binds to the STAT3 promoter, thus inhibiting the Th17 cell-mediated inflammation associated with fibrosis. (**C**) The presence of estrogen and gut dysbiosis augments lung fibrosis through TGF-β1 expression, thus demonstrating that multiple factors contribute to lung fibrosis pathophysiology.

**Table 1 cells-12-00766-t001:** Demographic information of the sarcoidosis, IPF, and scleroderma patients and the healthy control subjects used in this study.

	Healthy Control	Sarcoidosis	IPF	Scleroderma
**Number**	25	31	45	11
**Gender (male, female)**	7, 18	11, 20	36, 9	5, 6
**Age years median (Minimum, Maximum)**	44 (23, 65)	50 (27, 72)	66 (50, 83)	60 (49, 86)
**Race**	12 AA, 13 C	16 AA, 15 C	45 C	7 C, 3 AA, 1 Asian

ILD: Interstitial Lung Disease; C: Caucasian; AA: African American; IPF: Idiopathic Pulmonary Fibrosis.

## Data Availability

All of the sequences obtained from the lung and gut microbiome analysis of germ-free ABSL-1 and ABSL-2 mice have been deposited into BioProject ID, Accession number PRJNA899808.

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
