# Peer review of "Low Gut Microbial Diversity Augments Estrogen-Driven Pulmonary Fibrosis in Female-Predominant Interstitial Lung Disease"

_cells, 2023, doi:10.3390/cells12050766_

Round 1

Reviewer 1 Report

The profibrotic cytokines such as IL-17A and TGF-β1 have been implicated in interstitial lung disease (ILD) pathogenesis, interactions between gut dysbiosis, gonadotrophic hormones and molecular mediators of profibrotic cytokine expression, such as phosphorylation of STAT3, have not been defined.

Chromatin immunoprecipitation sequencing (ChIP-seq) analysis of primary human CD4+ T cells shown that regions within the STAT3 locus are significantly enriched for binding by the transcription factor estrogen receptor alpha (ERa). Genetic absence of ESR1 or ovariectomy in mice significantly increased pSTAT3 and IL-17A expression in pulmonary CD4+ T cells, which was reduced after repletion of female hormones. Assessment of lung fibrosis among menstruating females in different rearing environments revealed that environments favoring gut dysbiosis augment fibrosis. Hormone repletion following ovariectomy further augmented lung fibrosis, suggesting pathologic interactions between gonadal hormones and gut microbiota on lung fibrosis severity. Analysis in female sarcoidosis patients revealed a significant reduction in pSTAT3 and IL-17A levels and a concomitant increase in TGF-β1 levels in CD4+ T cells, compared to male sarcoidosis patients.

These studies reveal that estrogen is profibrotic in females and that gut dysbiosis in menstruating females augments lung fibrosis severity, supporting a critical interaction between gonadal hormones and gut flora in lung fibrosis pathogenesis.

It is interesting topic with a critical interaction between gonadal hormones and gut flora in lung fibrosis pathogenesis. However, there still have some issues need to be revised.

1.IL-17 is an inflammatory cytokine produced mainly by activated T cells, which can promote the activation of T cells and stimulate epithelial cells, endothelial cells, and fibroblasts. It can produce a variety of cytokines such as IL-6, IL-8, GM-CSF, CAM-1, thus leading to inflammation. The marker method and isolate method about Th17 cells should be provided.

2.STAT3 expression is critical for lung fibrosis. The experiment should consider the reference about link of STAT3 and inflammation (Consumption of the fish oil high-fat diet uncouples obesity and mammary tumor growth through induction of reactive oxygen species in pro-tumor macrophages. Cancer Research, 2020, 80(12): 2564-2574.).

3.The relationship between inflammatory factors and intestinal flora needs to be discussed(Whole grain benefit: oat β-glucan and phenolic compounds synergistically regulates hyperlipidemia via gut microbiota in high-fat-diet mice. Food & Function, 2022, 13(24), 12686-12696. Doi: 10.1039/d2fo01746f. Oat phenolic compounds regulate metabolic syndrome in high fat diet-fed mice via gut microbiota. Food Bioscience. 50(2022)101946. Doi: 10.1016/j.fbio.2022.101946).

4. The references should be update.

5. Please include any additional comments on the mechanism Figures.

Author Response

Reviewer 1

The profibrotic cytokines such as IL-17A and TGF-β1 have been implicated in interstitial lung disease (ILD) pathogenesis, interactions between gut dysbiosis, gonadotrophic hormones and molecular mediators of profibrotic cytokine expression, such as phosphorylation of STAT3, have not been defined.

Chromatin immunoprecipitation sequencing (ChIP-seq) analysis of primary human CD4+ T cells shown that regions within the STAT3 locus are significantly enriched for binding by the transcription factor estrogen receptor alpha (ERa). Genetic absence of ESR1 or ovariectomy in mice significantly increased pSTAT3 and IL-17A expression in pulmonary CD4+ T cells, which was reduced after repletion of female hormones. Assessment of lung fibrosis among menstruating females in different rearing environments revealed that environments favoring gut dysbiosis augment fibrosis. Hormone repletion following ovariectomy further augmented lung fibrosis, suggesting pathologic interactions between gonadal hormones and gut microbiota on lung fibrosis severity. Analysis in female sarcoidosis patients revealed a significant reduction in pSTAT3 and IL-17A levels and a concomitant increase in TGF-β1 levels in CD4+ T cells, compared to male sarcoidosis patients.

These studies reveal that estrogen is profibrotic in females and that gut dysbiosis in menstruating females augments lung fibrosis severity, supporting a critical interaction between gonadal hormones and gut flora in lung fibrosis pathogenesis.

It is interesting topic with a critical interaction between gonadal hormones and gut flora in lung fibrosis pathogenesis. However, there still have some issues need to be revised.

1.IL-17 is an inflammatory cytokine produced mainly by activated T cells, which can promote the activation of T cells and stimulate epithelial cells, endothelial cells, and fibroblasts. It can produce a variety of cytokines such as IL-6, IL-8, GM-CSF, CAM-1, thus leading to inflammation. The marker method and isolate method about Th17 cells should be provided.

Thank you for this comment. Th17 cells were identified by flow cytometric assessment for the transcription factor STAT3, and the cytokine, IL-17A (Figure 2, 3).  This is the most rigorous definition and widely accepted by scientists within the immunology community.  We apologize that this was not obvious.  We have included the methods and reference for defining Th17 cells by flow cytometry and have now added addition clarification in the methods section (page 7, paragraph 1).

2.STAT3 expression is critical for lung fibrosis. The experiment should consider the reference about link of STAT3 and inflammation (Consumption of the fish oil high-fat diet uncouples obesity and mammary tumor growth through induction of reactive oxygen species in pro-tumor macrophages. Cancer Research, 2020, 80(12): 2564-2574.).

Thank you; this is a helpful suggestion.  We have added that the influence of diet on host immunity should be considered for future studies and cited this reference (page 23, paragraph 1).

3.The relationship between inflammatory factors and intestinal flora needs to be discussed (Whole grain benefit: oat β-glucan and phenolic compounds synergistically regulates hyperlipidemia via gut microbiota in high-fat-diet mice. Food & Function, 2022, 13(24), 12686-12696. Doi: 10.1039/d2fo01746f. Oat phenolic compounds regulate metabolic syndrome in high fat diet-fed mice via gut microbiota. Food Bioscience. 50(2022)101946. Doi: 10.1016/j.fbio.2022.101946).

Thank you; this is another helpful suggestion.  We have added that the influence of diet on gut microbiota and metabolomic syndromes should be considered for future studies and cited both references (page 23, paragraph 2).

  1. The references should be update.

We have reviewed the references; the majority are recent.  If a reference is of a later date, the ones chosen are the original investigation, which is what the immunology scientific field expects to be cited.

  1. Please include any additional comments on the mechanism Figures.

Thank you for this suggestion.  We have added additional comments to the graphic abstract which outlines the mechanism (Page 24, paragraph 2).

Reviewer 2 Report

In this work Authors reported a possible cross-talk between gonadal hormones and gut microbiota in the development of lung fibrosis.The exploration of the sex-steroid signaling and  their mechanism in  the respiratory system could  open  new perspectives in the  prevention and treatment of lung diseases.

I would like to ask:

The estradiol in elderly males is comparatively high than menopausal females due to increased aromatase activity with age.

Is there a possible different evolution of the lung pathology in this specific age group ? What about the role of the gut microbiota?

As far as the symptomatology of the lung fibrosis, it has been reported that women are more symptomatic and show a worse quality of life compared to men. 

What is in your opinion the relationship about gut dysbiosis and clinical presentation of the pulmonary diseases in both sexes?

Thank you

Author Response

In this work Authors reported a possible cross-talk between gonadal hormones and gut microbiota in the development of lung fibrosis. The exploration of the sex-steroid signaling and their mechanism in the respiratory system could open new perspectives in the prevention and treatment of lung diseases.

I would like to ask:

The estradiol in elderly males is comparatively high than menopausal females due to increased aromatase activity with age.  Is there a possible different evolution of the lung pathology in this specific age group? What about the role of the gut microbiota?

Thank you for this intriguing comment. In Figure 6, we did not compare estradiol levels in elderly males to menopausal females. This data demonstrates estradiol levels between healthy control males (no lung fibrosis), males with scleroderma (milder lung fibrosis) and males with IPF (severe lung fibrosis).  Our murine data indicates that estrogen is also profibrotic in females and that gut dysbiosis in menstruating females augments lung fibrosis severity (Figure 4). We do not have data to determine if increased estradiol levels in elderly males is linked the role of gut microbiota in lung fibrosis severity.  This data supports a future investigation of the gut microbiota in humans with fibrotic lung disease is warranted.

As far as the symptomatology of the lung fibrosis, it has been reported that women are more symptomatic and show a worse quality of life compared to men.  What is in your opinion the relationship about gut dysbiosis and clinical presentation of the pulmonary diseases in both sexes?

Thank you for this interesting question.  There is sexual variation in the severity of various fibrotic lung diseases.  For example, with IPF-related lung fibrosis, male morbidity and mortality is significantly worse than females. We provide references 6-8 in the manuscript to support this observation. The focus of this work in the interaction of gut dysbiosis, estrogen and acute lung injury on lung fibrosis in females. .  In the conclusion, we note that gut dysbiosis and estrogen augment TGF-β1 mediated lung fibrosis, which is also associated with less severity than IL-17A-mediated pathology (Pg 25, paragraph 1).  Future independent investigation of interactions between gut dysbiosis and testosterone on male lung fibrosis severity is warranted.  This study will provide the foundational mechanistic knowledge to propel future studies that focus on gut dysbiosis in men and women with pulmonary fibrosis, but determining this association was beyond the scope of the current article. 

Thank you

Reviewer 3 Report

The original article by Chioma et al. investigates the role of estrogen in the development of pulmonary fibrosis and trying to explain the relation between sex hormones and gut microbiota diversity for the disease outcome. The work will be of great interest for basic and translational scientists. However, in order to be consider for publication several questions and comments should be addressed.

How author can explain the discrepancy between the finding that female hormones are pro-fibrotic, but also repress the inflammatory process. Moreover, females have better disease prognosis compared to males, and fibrotic process is more common in older age (post menopause) comparing to younger (menstruating females)?

Authors should provide brief description of the diversity of gut microbiota in mice housed in ABSL-1 and 2. 

Is the relation between sex hormones and gut microbiota direct or indirect? Authors should provide more evidence. A schematic presentation of the mechanism could be of a great help for understanding.

Authors provide too many figures which complicate the readability. Some of the figures could be moved to the supplementary files. 

Author Response

The original article by Chioma et al. investigates the role of estrogen in the development of pulmonary fibrosis and trying to explain the relation between sex hormones and gut microbiota diversity for the disease outcome. The work will be of great interest for basic and translational scientists. However, in order to be consider for publication several questions and comments should be addressed.

How author can explain the discrepancy between the finding that female hormones are pro-fibrotic, but also repress the inflammatory process. Moreover, females have better disease prognosis compared to males, and fibrotic process is more common in older age (post menopause) comparing to younger (menstruating females)?

Thank you for pointing this out; we now provide additional clarification. This clinical observation is difficult to appreciate unless one appreciates the role of estrogen and the gut microbiome. TGF-β1 is the master regulator of fibrosis. Figure 6 demonstrates that TGF-β1 is most predominant in female sarcoidosis patients. There is more fibrosis in females than males due to the increased TGF-β1 expression. In Figure 4, we see that gut dysbiosis augments lung fibrosis. When IL-6 induction occurs by gut microbiota, downstream signaling can lead to either IL-17A or TGF-B1 expression. IL-17A expression leads to pulmonary inflammation. Estrogen signaling provides protection against proinflammatory fibrosis due to the capacity of the ERα to bind to the STAT3 promoter (Figure 1). This reduction in lung inflammation improves prognosis. In menopausal females, the gut microbiome continues to drive lung fibrosis, but due to the reduced estrogen state, there is no inhibition of STAT3 expression and Th17 cell development; lung fibrosis can now be mediated by IL-17A which likely explains the increased symptoms after menopause. This explanation is now provided (Page 22 and 23, paragraph 1).

Authors should provide brief description of the diversity of gut microbiota in mice housed in ABSL-1 and 2.

Thank you for this suggestion. A detailed description of the gut microbiota is provided in the cited references in Communications Biology manuscript. We now provide a brief, detailed description of the flora within ABSL-1 and ABSL-2 environments, with the appropriate cited reference (Page 15, paragraph 1).

Is the relation between sex hormones and gut microbiota direct or indirect? Authors should provide more evidence. A schematic presentation of the mechanism could be of a great help for understanding.

This is an interesting comment. There are several considerations that must be addressed because as cited in the Discussion, the gut microbiota could possibly modulate estrogen levels through β glucuronidase activity. As the reviewer mentions, the hormones may impact the gut microbiota. For this to be investigated in a manner that will be appreciated by the scientific community, it would require an independent and expensive investigation that would likely require 12-18 months to conduct. We now add verbiage that reflects a more in-depth assessment of the interplay of gonadal hormones and gut dysbiosis on lung fibrosis pathogenesis, which warrants future investigations (Page 23, paragraph 1). We agree with you that there are enough figures in this paper. We now provide schematic presentation to aid understanding (Page 24).

Authors provide too many figures which complicate the readability. Some of the figures could be moved to the supplementary files.

We agree that the work is detailed, which was necessary in order to provide a rigorous investigation of the hypothesis. The subpanels in each figure were included in anticipation of the data that the scientific community will want in order to demonstrate scientific rigor. For example, Figures 2-3 outline each molecular mediator of Th17 cell synthesis, in order to identify the step at which estrogen signaling exerts its influence. In a similar manner, Figure 4 demonstrates a reduction in collagen content by two complementary experiments, which the scientific community will want to see within the figure. This is what is expected in other high-impact journals, akin to yours. Figure 5 has only five subpanels, each of which is necessary to compare the impact of gut microbiota by sex. Figure 6 entails molecular assessment of cells derived from human with interstitial lung disease, going through each molecular step associated with profibrotic cytokine expression, akin to Figures 2 and 3. Figure 6 also contains complementary assessments such as RT-PCR and flow cytometry, which the scientific field will expect in order to demonstrate rigor. We appreciate your insight and respectfully request that the subpanels for each figure remain.

Round 2

Reviewer 1 Report

The author has response to reviewer's comments point by point. It can be acceptable in current revision.

Author Response

Thank you for your comment that the manuscript can be accepted in the Round 2 revision.

Reviewer 3 Report

Could authors provide a track changes version of the manuscript? 

Author Response

Could authors provide a track changes version of the manuscript? 

Yes.  We apologize for not attaching the track changes version earlier with the R2 submission.  That was a mistake.  We are submitting it now.

Round 3

Reviewer 3 Report

-

Author Response

Dear Erin,

   I do not see any comments from Reviewer 3.  The paper has undergone professional editing from your editing service.  All requested changes were incorporated into the version 2 of the manuscript.  I am not sure what else needs to be done.  Are you able to send me specific comments?